# Time-Dependent Changes in Hematoma Expansion Rate after Supratentorial Intracerebral Hemorrhage and Its Relationship with Neurological Deterioration and Functional Outcome

**DOI:** 10.3390/diagnostics14030308

**Published:** 2024-01-31

**Authors:** Gaby Abou Karam, Min-Chiun Chen, Dorin Zeevi, Bendix C. Harms, Victor M. Torres-Lopez, Cyprien A. Rivier, Ajay Malhotra, Adam de Havenon, Guido J. Falcone, Kevin N. Sheth, Seyedmehdi Payabvash

**Affiliations:** 1Department of Radiology and Biomedical Imaging, Yale University School of Medicine, New Haven, CT 06520, USA; gaby.aboukaram@yale.edu (G.A.K.); min-chiun.chen@yale.edu (M.-C.C.); dorin.zeevi@yale.edu (D.Z.); bendix.harms@yale.edu (B.C.H.); ajay.malhotra@yale.edu (A.M.); 2Department of Neurology, Yale University School of Medicine, New Haven, CT 06520, USA; victor.torreslopez@yale.edu (V.M.T.-L.); cyprien.rivier@yale.edu (C.A.R.); adam.dehavenon@yale.edu (A.d.H.); guido.falcone@yale.edu (G.J.F.); kevin.sheth@yale.edu (K.N.S.); 3Center for Brain and Mind Health, Yale University School of Medicine, New Haven, CT 06520, USA

**Keywords:** intracerebral hemorrhage, hematoma expansion, neurological deterioration, hemorrhagic stroke, outcome

## Abstract

Background: Hematoma expansion (HE) following an intracerebral hemorrhage (ICH) is a modifiable risk factor and a treatment target. We examined the association of HE with neurological deterioration (ND), functional outcome, and mortality based on the time gap from onset to baseline CT. Methods: We included 567 consecutive patients with supratentorial ICH and baseline head CT within 24 h of onset. ND was defined as a ≥4-point increase on the NIH stroke scale (NIHSS) or a ≥2-point drop on the Glasgow coma scale. Poor outcome was defined as a modified Rankin score of 4 to 6 at 3-month follow-up. Results: The rate of HE was higher among those scanned within 3 h (124/304, 40.8%) versus 3 to 24 h post-ICH onset (53/263, 20.2%) (*p* < 0.001). However, HE was an independent predictor of ND (*p* < 0.001), poor outcome (*p* = 0.010), and mortality (*p* = 0.003) among those scanned within 3 h, as well as those scanned 3–24 h post-ICH (*p* = 0.043, *p* = 0.037, and *p* = 0.004, respectively). Also, in a subset of 180/567 (31.7%) patients presenting with mild symptoms (NIHSS ≤ 5), hematoma growth was an independent predictor of ND (*p* = 0.026), poor outcome (*p* = 0.037), and mortality (*p* = 0.027). Conclusion: Despite decreasing rates over time after ICH onset, HE remains an independent predictor of ND, functional outcome, and mortality among those presenting >3 h after onset or with mild symptoms.

## 1. Introduction

Spontaneous intracerebral hemorrhage (ICH) accounts for 10–15% of strokes, with an incidence rate of 24.6 per 100,000 person years [1,2]. However, with only about 20% of patients regaining functional independence and an approximately 40% rate of mortality at 1-month follow-up, ICH remains a major cause of disability and death worldwide [1,2]. Within the first few hours after ICH onset, cerebral bleeding tends to continue in a large proportion of patients, leading to the progressive accumulation of blood within the brain tissue and worsening of brain injury. While hematoma size and location, as key variables influencing ICH outcome, are not modifiable at the time of presentation, hematoma expansion (HE) is an independent predictor of poor outcome and mortality, with clinical evidence suggesting that it may be preventable [3]. Thus, given its prognostic significance and potentially modifiable nature, HE has emerged as an attractive therapeutic target in ICH patients. However, cumulative evidence shows that the rate of HE tends to decline with an increasing time gap from onset to baseline head CT, suggesting that HE rate decreases over time after ICH onset.

Clinically, it is challenging to assess early HE after ICH due to variable time gaps from symptom onset to the admission diagnostic brain scan. Early HE is reported in 18% to 38% of patients scanned within 3 h of ICH onset [4]. However, HE rates are lower among patients scanned in later hours after ICH symptom onset, with less than 20% of patients scanned beyond 6 h of symptom onset reported to experience HE after the initial scan [4]. Brott et al. reported a 26% rate of >33% hematoma growth among ICH patients scanned within the first hour after the initial CT scan, with an additional 12% hematoma growth between the baseline and follow-up scans performed 1 to 20 h after the baseline scan [5]. Although differences in reported rates of HE are, in part, due to variable definitions between studies [6], the majority of HE appears to occur early after ICH onset, as has been recently shown in an individual patient-level meta-analysis [7]. However, it remains to be determined whether the prognostic correlates of HE would also change over time after ICH onset. Characterization of the temporal changes in the rate of HE and its clinical implications are crucial in guiding patients’ monitoring, treatment planning, transfer to tertiary care centers, and design of clinical trials aimed at limiting HE.

In addition, approximately one-third of ICH patients will experience neurological deterioration (ND) within hours to days after onset [8,9,10]. ND after ICH will independently increase the risk of death and functional dependency [8,9,10]. Indeed, ICH patients who remain stable during the first week after onset are most likely to survive the first year [11]. Older age, larger baseline ICH volume, HE, perihematomal edema, the presence of intraventricular hemorrhage (IVH), and the development of hydrocephalus increase the risk of ND [12]. Some investigators have further categorized ND into early (within the first 24 or 48 h post-ICH) versus delayed/late (occurring 1 to 7 days after the onset) [4,9,10,11]. Nevertheless, both early and delayed/late ND are associated with higher risks of disability and death [10]. Different groups have shown that HE is associated with both early and delayed/late ND [10,11,12,13]. However, there remains a knowledge gap regarding whether there is a difference in the association of HE with ND as the rate of hematoma growth decreases over time after the ICH onset.

In this study, we determined the rate of HE over time, within the first 24 h from the ICH onset, among consecutive cohorts of patients with supratentorial ICH. Then, we examined the association of HE with ND, functional outcome, and mortality among those with hyperacute HE versus those with HE in later hours after ICH onset, correcting for other predictors of outcome in a multivariable analysis. Using causal mediation analysis, we also evaluated whether the effects of HE on outcome and mortality are mediated by ND. Finally, we examined the predictors of ND, clinical outcome, and mortality in subset of ICH patients who presented with mild symptoms.

## 2. Methods

### 2.1. Patients

We retrieved the clinical and imaging information of consecutive patients with ICH presenting to our center from 1 October 2014 to 30 September 2023 using prospectively collected data in the Yale Acute Brain Injury Biorepository [14]. Additional information was retrieved though a retrospective review of electronic medical records. We included patients with primary spontaneous supratentorial ICH who underwent the baseline head CT scan within the first 24 h after the symptom onset or last-known-well, with a follow-up head CT within 48 h of ICH symptom onset or last-known-well. The majority of patients with spontaneous ICH admitted to our center routinely undergo follow-up CT scans at 6 to 12 and 24 to 48 h intervals; however, any follow-up imaging is affected by each individual patient’s risk factor, clinical course, and treatment plan. In patients with multiple follow-up head CTs during the 48 h period, we chose the scan acquired closest to the 24 h post-admission head CT timepoint. In patients who underwent a neurosurgical procedure within the first 48 h after admission, we used the last pre-procedure head CT to determine follow-up hematoma volume. Patients with residual functional deficit from prior ICH, ischemic stroke, or other morbidities affecting baseline functional status (modified Rankin Scale, mRS > 2) were excluded from our analysis. We also excluded patients with artifact on admission or follow-up CT scans that interfered with the accurate delineation of ICH contours. Those with missing clinical information affecting accurate determination of ND or outcome were excluded, too. This study received approval from the institutional review board, and the individual patient consent requirement was waived due to the retrospective nature of our analysis.

### 2.2. Measurement of Hematoma Volumes

Head CT scans were anonymized and converted to Nifti format images. Trained research associates who were blinded to patients’ information and clinical history independently performed manual segmentation of hemorrhagic lesions on each head CT scan using 3D Slicer platform (https://www.slicer.org/), as described previously [15,16,17]. Prior to segmentation, research associates underwent a training session to ensure consistency and accuracy in hematoma segmentation. This training involved reviewing a set of sample CT scans with known ICH, discussing segmentation criteria, and practicing on a set of sample head CTs until a satisfactory level of reliability was achieved. The segmentation process involved outlining intra-parenchymal hemorrhagic lesions on each axial slice, with an intensity threshold of 40 to 200 Hounsfield units applied to facilitate segmentation [18,19]. In patients with multiple ICH lesions, the collective volume of all hemorrhagic lesions was calculated. All segmentations were reviewed by an experienced neuroradiologist (SP) for quality control and to determine if any additional edits were required.

### 2.3. Ascertainment of Imaging and Clinical Outcomes

The ICH volumes were calculated from manually segmented hematoma lesions on baseline and follow-up head CTs. According to the prior literature, HE was defined as a >6 mL or >33% increase in ICH volume on a follow-up scan compared to the baseline CT [20,21]. Neurological deterioration (ND) was defined as a ≥4-point increase on the NIH stroke scale (NIHSS) or a ≥2-point drop on the Glasgow coma scale (GCS) during the first week after admission [8,22,23]. Some investigators have further categorized ND into early (within the first 24 or 48 h) versus delayed/late (1 to 7 days post-ICH) [8,22,24]; however, because the post-admission NIHSS and GCS scores of our patients were acquired at variable timepoints after admission, we opted to evaluate ND during the first week. For clinical outcome, we used the follow-up mRS ascertained at 3-month follow-up or the closest interval available. For short-term outcome, we used the discharge mRS. Poor outcome was defined as 4 to 6 mRS, as is commonly applied for ICH patients [22].

### 2.4. Statistical Analysis

The data are presented as frequency (percentage), mean ± standard deviation (SD), median (interquartile), or odds ratio (95% confidence interval). For a univariable analysis of nominal variable distributions, we used the Chi square test, for comparing continuous variable means, we used the student *t* test, and for comparing the distribution of ordinal variables (i.e., NIHSS, GCS, and mRS), we used the Wilcoxon rank sum test. For multivariable analysis, we used binary logistic regression with stepwise forward (*p* = 0.05) and backward (*p* = 0.10) variable selection and standardized all continuous independent variables by centering each feature around the mean with a unit SD. Only variables with significant differences in univariable analysis were included. We also applied causal mediation analysis to determine whether ND mediates or explains the relationship between HE with poor outcome and mortality. For mediation analysis, we fit multivariable probit regressions using a general linear model to determine the direct and indirect effects of HE on outcome and mortality, considering ND as a potential mediator and correcting for other clinical and imaging predictors of outcome and mortality. Statistical analysis was performed using STATA version 16. For causal mediation analysis, we used the “mediation” R package version 4.1.

## 3. Results

### 3.1. Patients Characteristics

A total of 567 patients were included in our analysis. Figure 1 depicts the flowchart of patients’ inclusion in our study. The mean age of patients was 69.8 ± 14.2 years, with 308 (54.3%) being male. The median admission NIHSS was 10 (4–18). None of the patients had bilateral acute ICH. In Appendix A, we compared the characteristics of patients who were included in our analysis versus those with supratentorial ICH and follow-up CT who were excluded due to a baseline CT obtained after the first 24 h from onset or a follow-up CT obtained more than 48 h from onset. The 43 patients who were excluded from the analysis (fulfilling the above criteria) had lower systolic blood pressure, less severe symptoms (lower admission NIHSS), and higher blood glucose levels at the time of admission with better clinical outcome (lower 3-month mRS) compared to the 567 patients who were included in our analysis. They were also more likely to be white and have deep ICH (Appendix A).

### 3.2. Rates of HE Based on the Time from Onset to Baseline Scan

In our cohort, 177 (31.2%) of patients experienced HE, comparing ICH volume on baseline versus follow-up head CT. Table 1 summarizes the clinical characteristics, hematoma volumes, and outcome among patients with and without HE. Overall, HE was more common among those scanned earlier after the onset; these patients also had larger baseline hematomas, lobar ICH, and worse clinical symptoms at presentation—i.e., higher NIHSS and lower GCS (Table 1). Notably, the rate of hypertension was higher among those without HE compared to those with HE. Patients with HE subsequently had higher rates of ND, 3-month poor outcome, and mortality, as well as surgical evacuation (Table 1). Follow-up hematoma volumes were expectedly larger among those with HE compared to those without. Figure 2A summarizes the rates of HE among patients scanned over the first 24 h after ICH onset. Overall, HE rates were higher among those scanned earlier, with 124 (40.8%) of 304 scanned within the first 3 h showing HE versus 53 (20.2%) of 263 who were scanned 3 to 24 h after ICH onset showing HE (*p* < 0.001). Notably, the relationship between the rate of HE and the time gap from onset to baseline CT (Figure 3A) over time after ICH onset can be explained by either a linear (fitted R square = 0.51) or logarithmic regression model (fitted R square = 0.47). In subgroup analysis, in both cohorts of 304 patients scanned within the first 3 h, as well as 263 who were scanned 3 to 24 h after ICH onset, baseline hematoma volumes were the only independent predictors of HE (both *p* < 0.001).

### 3.3. Rates of ND and Its Association with HE Based on the Time from Onset to Baseline Scan

In our cohort, 212 (37.4%) of patients experienced ND during the first week of admission. Table 2 summarizes the clinical characteristics, hematoma volumes, and outcome among patients with and without ND. Overall, ND was more common among those who presented at an older age, were scanned earlier, and had worse symptoms at presentation (higher NIHSS). In addition, patients with ND had larger baseline and follow-up volumes and higher rates of HE, as well as higher rates of intraventricular hemorrhage (Table 2). These patients had worse functional outcomes at the time of discharge and higher rates of 3-month poor outcome and mortality (Table 2). Figure 2B and Figure 3B summarize the rates of HE and ND among patients scanned in the first 24 h after ICH onset. There was a higher number of patients with HE and subsequent ND among those scanned earlier after ICH onset (Figure 2B). The rate of ND among those with HE who were scanned within the first 3 h was 54.0% (67/124) (OR = 2.6 (1.6–4.2) (*p* < 0.001)), and the rate of ND among those with HE who were scanned 3 to 24 h after the ICH onset was 45.2% (23/53) (OR = 1.7 (1.1–3.1) (*p* = 0.047)). Notably, the relationship between the rate of HE with ND and the time gap from onset to baseline CT (Figure 3B) can be explained by either a linear (fitted R square = 0.42) or logarithmic (fitted R square = 0.34) regression model.

### 3.4. Odds of ND, Poor Outcome, and Mortality Associated with HE

Table 3 summarizes the clinical characteristics of patients with favorable versus poor outcomes at 3-month follow-up, and Table 4 summarizes the clinical characteristics of patients who survived versus those who are deceased. Given that patients with HE had larger hematomas and worse clinical symptoms at presentation, we evaluated the association of HE with ND, poor outcome, and mortality after correcting for patients’ age, admission NIHSS, GCS, baseline hematoma volume, intraventricular hemorrhage, external ventricular drainage placement, surgical evacuation, and history of hypertension in multivariable logistic regression. Among patients scanned within the first 3 h of ICH onset, HE was an independent predictor of ND (OR = 2.7 (1.6–4.5) (*p* < 0.001)), poor outcome (OR = 2.4 (1.2–4.6) (*p* = 0.010)), and mortality (OR = 2.7 (1.4–5.2) (*p* = 0.003)), after correcting for other risk factors and therapeutic interventions. Similarly, among patients scanned 3 to 24 h after ICH onset, HE was an independent predictor of ND (OR = 1.4 (1.0–2.4) (*p* = 0.043)), poor outcome (OR = 1.7 (1.1–3.7) (*p* = 0.037)), and poor outcome and mortality (OR = 3.0 (1.4–6.3) (*p* = 0.004)), after correcting for other risk factors and therapeutic interventions. In mediation, the analysis adjusted for admission NIHSS, GCS, age, baseline hematoma volume, and presence IVH, as well as external ventricular drain placement and surgical evacuation. The effects of HE on poor outcome and death was mediated by ND, with average causal mediation effects of 0.03 (0.01–0.05), and 0.03 (0.01–0.05), respectively (both *p* < 0.001).

### 3.5. Predictors of ND, Poor Outcome, and Mortality among ICH Patients with Mild Presentation

In our dataset, 180/567 (31.7%) patients had mild symptoms at presentation, i.e., admission NIHSS ≤ 5. However, among these 180 patients, 53 (29.4%) experienced ND, 36 (20%) had poor 3-month outcomes, and 20 (11%) died during the follow-up period. Notably, the rates of poor outcome at 3-month follow-up were 15% to 25% among ICH patients with mild admission symptoms scanned over the first 24 h (Figure 4). In a multivariable model including admission NIHSS, GCS, age, baseline hematoma volume, HE, history of hypertension, surgical evacuation, and external ventricular drainage, we found that independent predictors of ND were admission NIHSS (OR = 1.3 (1.1–1.6) (*p* = 0.008)) and presence of the IVH (OR = 1.9 (1.0–3.6) (*p* = 0.049)). Independent predictors of poor outcome were age (OR = 1.1 (1.0–3.0) (*p* = 0.003)) and admission NIHSS (OR = 1.3 (1.0–1.9) (*p* = 0.018)). Independent predictors of mortality were age (OR = 1.1 (1.0–1.1) (*p* = 0.046)), admission NIHSS (OR = 1.3 (1.0–1.7) (*p* = 0.013)), and IVH (OR = 2.6 (1.0–6.3) (*p* = 0.049)). Although HE—defined as >6 mL or >33% growth and affecting 35/180 (19.4%) individuals—was not an independent predictor of ND, poor outcome, or mortality among ICH patients with mild symptoms at admission, we found that a >6 mL hematoma growth—affecting 11/180 (6.1%) individuals—was an independent predictor of ND (OR = 4.1 (1.0–16.6, *p* = 0.026)), poor outcome (OR = 3.1 (1.1–10.3, *p* = 0.037)), and mortality (OR = 4.1 (1.1–14.6, *p* = 0.027)) in this subgroup of ICH patients.

## 4. Discussion

In this study, we demonstrated that the rate of subsequent HE is higher among those scanned closer to ICH onset compared to those scanned later. This suggests that hematoma growth tends to occur in the early hours following ICH onset and then slows down. Although the rate of HE declines among those scanned later (e.g., 3 to 24 h after onset), HE remains an independent predictor of ND, poor outcome, and mortality. The effects of HE on poor outcome and mortality was, in part, mediated by ND. Also, among a subset of ICH patients with mild symptoms (admission NIHSS ≤ 5), hematoma growth exceeding 6 mL was an independent predictor of ND, poor outcome, and mortality. Our findings underscore the importance of identifying patients at risk of HE for potential anti-expansion or hemostatic treatment, even when admitted >3 h after the ICH onset or presenting with mild symptoms.

These findings confirm prior studies suggesting that HE tends to occur in the very first hours after the ICH onset, although it is still possible that HE occurs later on, up to 24 h after the onset [25,26]. Prior studies have also shown that HE is more common among patients scanned earlier after ICH onset than those scanned later [4,5,6,7]. Kazui et al. showed that the frequency of HE progressively declined as the time from onset to the initial scan increases at a rate of 36% in patients scanned ≤3 h from the onset, 16% among those scanned 3 to 6 h post-onset, 15% for those scanned 6 to 12 h post-onset, 6% for those scanned 12 to 24 h post-onset, and 0% for those scanned 24 to 48 h post-onset [25]. In a recent meta-analysis, the probability of ICH growth, with increasing time from symptom onset to baseline imaging, decreased in a nonlinear fashion [7]. The rate of the decline in HE rate was steepest from 0.5 to 3 h after ICH onset [7]. Some investigators attribute the time course of changes in HE rate to the dynamic process of ICH with intermittent bleeding episodes [26]. They hypothesize that the initial vessel rupture and ICH would trigger the secondary rupture of nearby vessels exposed to mechanical pressure and ischemia from the hematoma’s mass effect [26,27]. Secondary vessel rupture and intermittent bleeding episodes may account for plateau phases, wherein the hematoma volume remains constant between active expansion episodes [25,26,28,29]. However, in the absence of serial imaging in all patients, it is challenging to determine whether HE among those scanned >3 h after the onset was due to persistent growth following the hemorrhage onset or an intermittent expansion process intervened with a stable plateau in hematoma volume. Regardless, cumulative evidence, including our results, suggests that over a third of spontaneous ICH tends to expand within the first three hours, with the expansion rate gradually slowing down as time progresses from the onset.

The higher rates of HE in the earliest hours after the onset of ICH underscore the imperative for expedited randomization and trial interventions aimed at preventing hematoma growth. However, our findings, revealing poor prognostic correlates of HE among those scanned 3 to 24 h after the onset, suggest that limiting HE in high-risk ICH patients admitted >3 h after the onset can still be beneficial. Nevertheless, because a smaller proportion of patients will experience HE in later hours after the ICH onset, any potential benefit of anti-expansion therapy delivered to all patients in the subacute phase may be diluted. Thus, the identification of patients at risk of HE for targeted therapy is more relevant among those presenting 3 to 24 h after ICH onset.

We have also demonstrated that the impact of HE on poor outcome and mortality is mediated by ND. Even after adjusting for well-known predictors of mortality and poor outcome, ND remains a crucial independent predictor of functional outcome and mortality [11]. In fact, patients with ICH who maintain stability during the first week of admission are most likely to survive the first year [11]. Analysis of the tranexamic acid in intracerebral hemorrhage-2 (TICH-2) trial dataset revealed that both early (≤48 h) and late (48 h to 7 days after onset) ND were associated with a significantly higher risk of dependency and death, as well as worse disability, cognition, and quality of life [10]. Lord et al. reported the association of HE with early ND (1–24 h) [9], and Okazaki et al. reported that hematoma growth over the first 24 h was an independent predictor of late ND [13]. Lastly, Sorimachi and Fujii reported that even a 6% increase in ICH volume may result in ND [30]. Our findings further underscore the contribution of HE to ND, with ICH expansion being an independent predictor of ND among patients scanned both <3 h and 3 to 24 h after onset. Additionally, for the first time in the literature, we have shown that the adverse prognostic effects of HE on functional outcome and mortality are (partially) mediated by ND. These findings emphasize the importance of HE as a treatment target to prevent ND, leading to improved morbidity and mortality rates.

Finally, our investigation revealed that, among patients presenting with mild symptoms at the time of admission, typically defined by an admission NIHSS ≤ 5 [31,32], hematoma growth exceeding 6 mL serves as an independent predictor of ND, poor outcome, and death. Notably, individuals with mild symptoms at admission may still experience ND (29.4% in our series) and poor outcome (20% in our series), or even death (11%). Therefore, even those exhibiting mild neurological deficits at admission may benefit from anti-expansion therapies. Our findings suggest that preventing HE—whether in patients presenting more than three hours after ICH onset or with mild symptoms—has the potential to improve outcome and reduce mortality.

Our study has several strengths that contribute to its clinical relevance. Firstly, we leveraged a relatively large sample size from a tertiary care stroke center. This contributed to the statistical power of our findings and improved the generalizability of our results. The combination of prospective data registry and a retrospective review of medical records provided comprehensive information regarding the clinical course during admission and medium-term functional outcome of patients, allowing us to evaluate ND and 3-month outcome. Additionally, we conducted mediation analysis to determine the underlying mechanisms influencing the relationship between HE and ND with morbidity and mortality. Finally, our multivariable analyses were adjusted for other baseline predictors of outcome and death among ICH patients, as well as therapeutic interventions, including surgical evacuation and external ventricular drainage.

The primary limitation of our study is the absence of serial imaging in all patients, preventing us from confidently distinguishing between those with continuous versus intermittent hematoma growth over time following ICH onset. Collecting such detailed information, however, is not practically feasible given that serial imaging may interfere with the intensive care required by most ICH patients after admission. Additionally, due to the lack of regular recording of post-admission NIHSS and GCS, we were unable to subcategorize ND into early versus late/delayed forms. Our inclusion and exclusion criteria also induce selection bias, as indicated in Appendix A. Patients with milder symptoms and lower blood pressure at presentation were less likely to undergo head CT scans within 24 h of ICH onset or obtain follow-up CT scans within 48 h post-onset and, thus, were excluded from our study. Our analysis has also inherently excluded patients with massive ICH who succumbed before any follow-up imaging was obtained. Thus, our selection criteria have inadvertently omitted individuals exhibiting symptoms at the most extreme ends of the spectrum. Our analysis is also inherently limited to those with spontaneous supratentorial ICH. Moreover, because the majority of patients were admitted shortly after symptom onset, there was a relative scarcity of data for those scanned more than three hours post-ICH, rendering our cohort underpowered to further analyze patients in smaller time bracket bins, such as 3 to 6, 6 to 9, and 9 to 12 h post-ICH onset, etc. In addition, aside from surgical evacuation and external ventricular drainage, the details of medical management for blood pressure reduction and strategies for the reversal of anticoagulation could not be systematically organized and analyzed in this retrospective analysis, given the variability of therapeutic options in the absence of universal guidelines. Notably, none of the patients in our cohort received hemostatic agents. Moreover, although our cohort has a relatively large sample size, it could not represent some of the well-established risk-factors in ICH patients. For example, there was no significant difference in the systolic blood pressure of patients with and without HE, although higher blood pressure is a known risk-factor for HE. Finally, it should be noted that variable definitions of HE may result in different results. For example, the use of percentage versus absolute increase in volume for defining HE affects the reported rate of expansion based on the hemorrhage location [33].

## 5. Conclusions

Our results suggest that following ICH onset, cerebral hematomas tend to expand at much higher rates in the earliest hours, with the rate of HE gradually decreasing as time progresses. However, despite the decreasing rate of HE over time after presentation, HE remains an independent predictor of ND, poor outcome, and death. Even among patients presenting with mild symptoms, ICH growth is an independent risk factor for ND, poor outcome, and mortality. Notably, the effects of HE on outcome and death is partly mediated via the ensuing ND over the first week after admission. Overall, our findings underscore the importance of HE as a treatment target, even among ICH patients presenting >3 h after onset or with mild symptoms. However, given that the rate of HE is lower among those admitted and scanned >3 h after symptom onset, the identification of those at risk of HE is more relevant in this population to detect any treatment benefit from targeted anti-expansion therapies. 

## Figures and Tables

**Figure 1 diagnostics-14-00308-f001:**
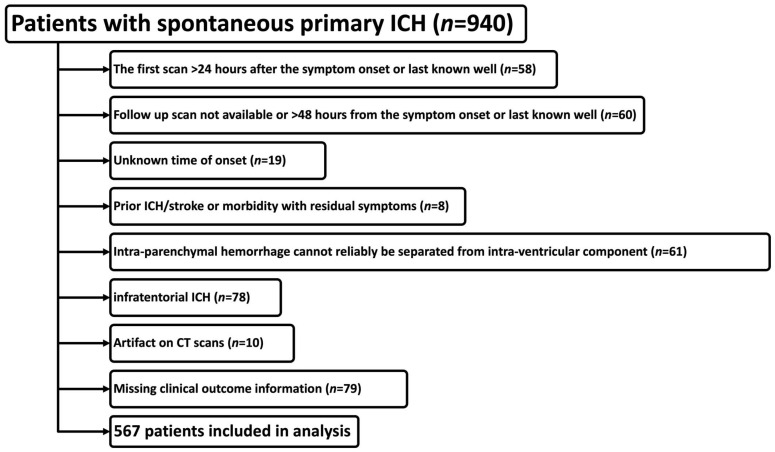
Flowchart of patient inclusion and exclusion in our analysis. The lines and boxes depict the stepwise exclusion criteria and the corresponding number of patients excluded at each step.

**Figure 2 diagnostics-14-00308-f002:**
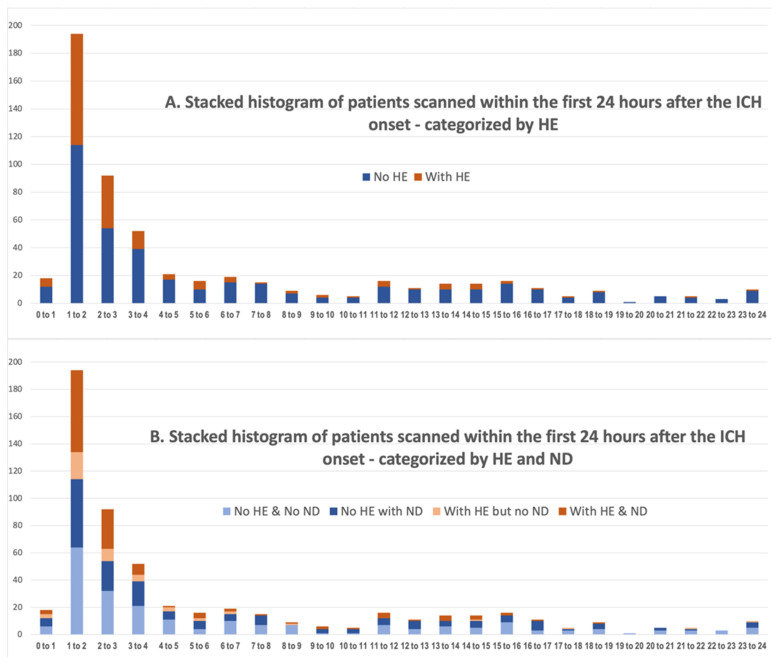
Stacked histogram of patients scanned within the first 24 h after the intracerebral hemorrhage (ICH) onset, categorized by those who developed (**A**) Hematoma expansion (HE) and (**B**) HE and/or neurological deterioratin (ND).

**Figure 3 diagnostics-14-00308-f003:**
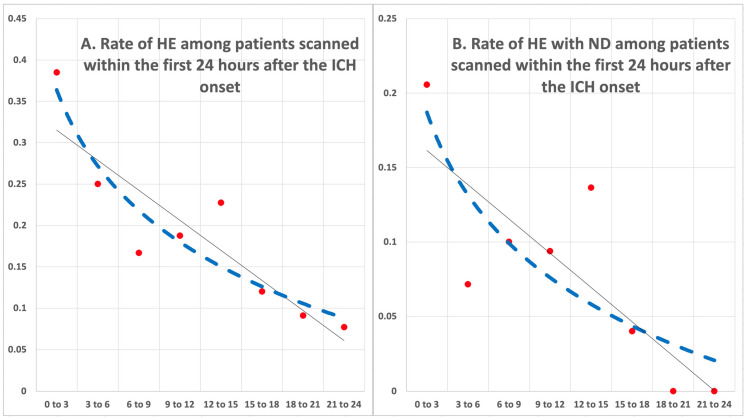
(**A**) Rate of hematoma expansion (HE) and (**B**) HE associated with neurological deterioration (ND) over the first 24 h after intracerebral hemorrhage (ICH).

**Figure 4 diagnostics-14-00308-f004:**
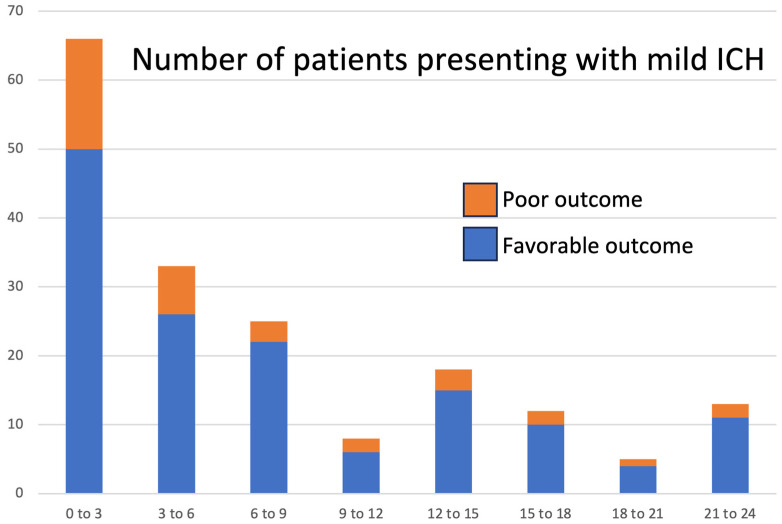
Stacked histogram of patients scanned within the first 24 h after intracerebral hemorrhage (ICH) onset presenting with mild symptom severity (NIHSS ≤ 5) categorized by poor (mRS 4 to 6) versus favorable (mRS 0 to 3) outcome at 3-month follow-up.

**Table 1 diagnostics-14-00308-t001:** Clinical characteristics of patients with and without hematoma expansion (HE).

	Without HE (*n* = 390)	With HE (*n* = 177)	*p* Value
Male sex	205 (52.6%)	103 (58.2%)	0.237
Age (years)	69.6 ± 13.9	70.3 ± 14.7	0.546
Ethnicity—Hispanic	32 (8.2%)	14 (7.9%)	1.00
Race—			0.532
White	262 (67.2%)	129 (72.9%)	
Black	116 (29.7%)	44 (24.9%)	
Asian	11 (2.8%)	4 (2.3%)	
Other	1 (0.3%)	0 (0%)	
Systolic blood pressure (mmHg)	172.8 ± 31.2	170.6 ± 33.2	0.447
History of hypertension	339 (86.9%)	139 (78.5%)	0.013
Baseline blood glucose (mg/dL)	146.4 ± 65.6	146.4 ± 49.4	0.086
History of anticoagulation	71 (18.2%)	44 (24.9%)	0.072
Onset to baseline CT time gap (hours)	6.1 ± 6.3	3.4 ± 4.5	<0.001
Baseline hematoma volume (mL)	18.1 ± 20.4	24.9 ± 23.5	0.001
Baseline to follow-up CT time gap (hours)	17.9 ± 17.2	19.2 ± 13.9	0.388
Follow-up hematoma volume (mL)	18.5 ± 19.4	42.5 ± 33.7	<0.001
Admission Glasgow coma scale score	14 (11–15)	13 (8–15)	0.001
Admission NIH stroke scale score	8 (4–17)	14 (7–20)	<0.001
Deep (vs. lobar) hemorrhage	244 (62.6%)	88 (49.7%)	0.004
Intraventicular hemorrhage	192 (49.2%)	98 (55.4%)	0.204
External ventricular drainage	52 (13.3%)	14 (7.9%)	0.067
Surgical evacuation	13 (3.3%)	16 (9.0%)	0.007
Neurological deterioration (ND)	122 (31.3%)	90 (50.8%)	<0.001
Discharge modified Rankin scale score (mRS)	4 (3–5)	5 (4–6)	<0.001
3-month follow-up (mRS)	3 (2–5)	5 (3–6)	<0.001
Poor outcome (mRS 4 to 6)	179 (45.9%)	127 (71.8%)	<0.001
Mortality	79 (20.3%)	80 (45.2%)	<0.001

**Table 2 diagnostics-14-00308-t002:** Clinical characteristics of patients with and without neurological deterioration (ND)—defined by a ≥4-point increase in NIHSS or a ≥2-point drop in GCS in the first week of admission.

	Without ND (*n* = 355)	With ND (*n* = 212)	*p* Value
Male sex	191 (53.8%)	117 (55.2%)	0.794
Age (years)	68.5 ± 14.7	72.0 ± 13.1	0.044
Ethnicity—Hispanic	31 (8.7%)	15 (7.1%)	0.528
Race—			0.141
White	237 (66.8%)	154 (72.6%)	
Black	110 (31.0%)	50 (23.6%)	
Asian	8 (2.3%)	7 (3.3%)	
Other	0 (0%)	1 (0.5%)	
Systolic blood pressure (mmHg)	171.8 ± 32.9	172.6 ± 30.1	0.139
History of hypertension	298 (83.9%)	180 (84.9%)	0.812
Baseline blood glucose (mg/dL)	143.3 ± 64.1	143.9 ± 56.0	0.087
History of anticoagulation	71 (20.0%)	44 (20.8%)	0.830
Onset to baseline CT time gap (hours)	5.5 ± 6.2	4.7 ± 5.5	0.032
Baseline hematoma volume (mL)	17.3 ± 20.3	25.0 ± 22.9	0.006
Baseline to follow-up CT time gap (hours)	17.0 ± 17.0	20.5 ± 14.7	0.689
Follow-up hematoma volume (mL)	20.3 ± 24.1	33.6 ± 30.2	<0.001
Admission Glasgow coma scale score	14 (10–15)	14 (10–15)	0.131
Admission NIH stroke scale score	9 (3–18)	12 (6–18)	0.013
Deep (vs. lobar) hemorrhage	215 (60.6%)	117 (55.2%)	0.218
Intraventicular hemorrhage	154 (43.4%)	136 (64.2%)	<0.001
External ventricular drainage	36 (10.1%)	30 (14.2%)	0.176
Surgical evacuation	17 (4.8%)	12 (5.7%)	0.695
Hematoma expansion (HE)	87 (24.5%)	90 (42.5%)	<0.001
Discharge modified Rankin scale score (mRS)	4 (3–5)	5 (4–6)	<0.001
3-month follow-up (mRS)	3 (2–5)	5 (3–6)	<0.001
Poor outcome (mRS 4 to 6)	158 (44.5%)	148 (69.8%)	<0.001
Mortality	74 (20.8%)	85 (40.1%)	<0.001

**Table 3 diagnostics-14-00308-t003:** Clinical characteristics of patients with favorable (modified Rankin score 0 to 3) versus poor (4 to 6) outcomes.

3-Month Outcome	Favorable (*n* = 261)	Poor (*n* = 306)	*p* Value
Male sex	146 (55.9%)	162 (52.9%)	0.499
Age (years)	67.7 ± 13.9	71.6 ± 14.2	0.001
Ethnicity—Hispanic	19 (7.3%)	27 (8.8%)	0.540
Race—			0.447
White	186 (71.3%)	205 (67.0%)	
Black	67 (25.7%)	93 (30.4%)	
Asian	7 (2.7%)	8 (2.6%)	
Other	0 (0%)	1 (0.2%)	
Systolic blood pressure (mmHg)	172.5 ± 30.1	171.8 ± 33.4	0.783
History of hypertension	219 (83.9%)	259 (84.6%)	0.818
Baseline blood glucose (mg/dL)	136.9 ± 58.8	149.1 ± 62.6	0.018
History of anticoagulation	39 (14.9%)	76 (24.8%)	0.005
Onset to baseline CT time gap (hours)	5.9 ± 6.3	4.7 ± 5.7	0.014
Baseline hematoma volume (mL)	13.0 ± 14.1	26.3 ± 24.8	<0.001
Baseline to follow-up CT time gap (hours)	17.6 ± 18.7	18.9 ± 13.8	0.330
Follow-up hematoma volume (mL)	14.1 ± 15.4	34.8 ± 31.4	<0.001
Admission Glasgow coma scale score	15 (14–15)	12 (7–14)	<0.001
Admission NIH stroke scale score	5 (2–10)	17 (10–22)	<0.001
Deep (vs. lobar) hemorrhage	146 (55.9%)	186 (60.8%)	0.266
Intraventicular hemorrhage	94 (36.0%)	196 (64.1%)	<0.001
External ventricular drainage	10 (3.8%)	56 (18.3%)	<0.001
Surgical evacuation	6 (2.3%)	23 (7.5%)	0.006
Hematoma expansion (HE)	50 (19.2%)	127 (41.5%)	<0.001
Discharge modified Rankin scale score (mRS)	3 (2–4)	5 (4–6)	<0.001
Neurological deterioration (ND)	64 (24.5%)	148 (48.4%)	<0.001

**Table 4 diagnostics-14-00308-t004:** Clinical characteristics of alive versus deceased patients at 3-month follow-up.

	Alive (*n* = 408)	Deceased (*n* = 159)	*p* Value
Male sex	219 (53.7%)	89 (56.0%)	0.640
Age (years)	68.2 ± 13.9	73.9 ± 14.2	<0.001
Ethnicity—Hispanic	34 (8.3%)	12 (7.5%)	0.865
Race—			0.463
White	275 (67.4%)	116 (73.0%)	
Black	122 (29.9%)	38 (23.9%)	
Asian	10 (2.5%)	5 (3.1%)	
Other	1 (0.2%)	0 (0%)	
Systolic blood pressure (mmHg)	172.2 ± 30.8	172.0 ± 34.5	0.960
History of hypertension	345 (84.6%)	133 (83.6%)	0.798
Baseline blood glucose (mg/dL)	139.7 ± 60.3	153.2 ± 62.5	0.018
History of anticoagulation	66 (16.2%)	49 (30.8%)	<0.001
Onset to baseline CT time gap (hours)	5.3 ± 5.9	5.0 ± 6.1	0.603
Baseline hematoma volume (mL)	15.0 ± 16.1	33.4 ± 27.6	<0.001
Baseline to follow-up CT time gap (hours)	18.9 ± 17.4	16.9 ± 12.9	0.206
Follow-up hematoma volume (mL)	17.4 ± 18.8	45.3 ± 34.7	<0.001
Admission Glasgow coma scale score	14 (12–15)	11 (6–14)	<0.001
Admission NIH stroke scale score	8 (3–14)	18 (11–24)	<0.001
Deep (vs. lobar) hemorrhage	246 (60.3%)	86 (54.1%)	0.185
Intraventicular hemorrhage	171 (41.9%)	119 (74.8%)	<0.001
External ventricular drainage	38 (9.3%)	28 (17.6%)	0.008
Surgical evacuation	15 (3.7%)	14 (8.8%)	0.018
Hematoma expansion (HE)	127 (31.1%)	85 (53.5%)	<0.001
Neurological deterioration (ND)	97 (23.8%)	80 (50.3%)	<0.001

## Data Availability

All data are available from the corresponding author upon reasonable request.

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
