# Peer review of "Time-Dependent Changes in Hematoma Expansion Rate after Supratentorial Intracerebral Hemorrhage and Its Relationship with Neurological Deterioration and Functional Outcome"

_diagnostics, 2024, doi:10.3390/diagnostics14030308_

Round 1

Reviewer 1 Report

Comments and Suggestions for Authors

This study investigated Hematoma expansion over time (within 48 h) and its association with  neurological deterioration (within a week) and clinical outcome at 3 months.

This is a retrospective study, which included supratentorial ICH patients who were admitted within 24 hours and re-scanned within 48 hours. Those who did not have follow-up CT within 48 hours were excluded. I wonder if there is imaging protocol who get the follow up CT witnih 48 hours. Do they routinely perform CT scan within 48 hours?

For attenuating the limitation of retrospective study design, it would be better that the comparison between the excluded patients due to no fu CT image within 48 hours and those included.

In their flow chart of patient enrollment, they should clarify the lines are showing the reasons for excluding patients.  

The asset of this study is a large number of subjects.  Number = 567 

In their conclusion, Hematoma expansion is frequent in patients with early CT scan. HE was related to ND, poor outcome and mortality. They also emphasized the early anti-expansion treatment.

I think this is helpful results worthy of sharing between physicians in their clinical practice.

Author Response

We thank the editor and reviewers for their careful reviews of our manuscript and insightful comments which helped improve our work. Below is an itemized list of reviewers’ comments and our corresponding response immediately follows. We have highlighted the changes in the revised manuscript and supplemental material documents.

Reviewer #1

C.1. I wonder if there is imaging protocol who get the follow up CT within 48 hours. Do they routinely perform CT scan within 48 hours?

R: In the revised methods (2.1. Patients section), we have clarified that “The majority of patients with spontaneous ICH admitted to our center routinely undergo follow-up CT scans at 6-to-12 and 24-to-48 hours intervals; however, any follow-up imaging is affected by each individual patient’s risk-factor, clinical course, and treatment plan.

C.2. For attenuating the limitation of retrospective study design, it would be better that the comparison between the excluded patients due to no fu CT image within 48 hours and those included.

R: We thank the reviewer for their insightful comments which helped us improve the manuscript. In the revised manuscript, we have compared the characteristics of patients who were included in our analysis with those who had supratentorial ICH but were excluded due to time-frame criteria. The information is added to Results (section 3.1. Patients Characteristics and supplemental Table): “In the Supplemental Table, we compared the characteristics of patients who were included in our analysis versus those with supratentorial ICH and follow-up CT who were excluded due to a baseline CT obtained after the first 24 hours from the onset or follow-up CT obtained greater than 48 hours from the onset. The 43 patients who were excluded from the analysis (fulfilling the above criteria) had lower systolic blood pressure, less severe symptoms (lower admission NIHSS), and higher blood glucose levels at the time of admission with better clinical outcome (lower 3-month mRS) compared to 567 patients who were including in our analysis. They were also more likely to be from white race and have deep ICH (supplemental Table).”

In the limitation of our study, we have also added that “Our inclusion and exclusion criteria induce selection bias as pointed out in the supplemental Table. Patients with milder symptoms and lower blood pressure at presentation were more likely not to undergo head CT scans within 24 hours from the ICH onset or obtain a follow-up CT within the 48-hour post-onset time window and thus were excluded from our study. Our analysis has also inherently excluded patients with massive ICH who succumbed before any follow-up imaging obtained. Thus, our selection criteria have inadvertently omitted individuals exhibiting symptoms at the most severe end of the spectrum.”

Q.3. In their flow chart of patient enrollment, they should clarify the lines are showing the reasons for excluding patients. 

R: We have revised the figure legend, and now it states that “Flowchart of patient inclusion and exclusion in our analysis. The lines and boxes depict the stepwise exclusion criteria and the corresponding number of patients excluded at each step.”

Reviewer 2 Report

Comments and Suggestions for Authors

In this study the authors aimed to investigate the association of hematoma expansion with neurological deterioration, functional outcome and mortality based on the time-gap from onset to baseline CT. Some concerns and suggestions are listed as below:

Why the follow-up head CT within 48 hours (or multiple follow-up head CTs) from the ICH symptom onset was done? Potential selection bias (hematoma expansion only in scanned earlier after the onset) should not be ignored.

Why systolic blood pressure was almost balanced in HE and non-HE groups (P=0.447)?

How these patients were treated in this study? Some may argue that different treatment may have effects on final results.

Regarding the rate of subsequent hematoma expansion, did you compare the data with other medical centers?

How about the relationship between hematoma expansion and short-term outcomes?

Which patients tend to expand at much higher rate in the earliest hours? How about changes of their blood pressure in this study?

How about the relationship between hematoma expansion and the localization of intracerebral hematoma? Additional analysis should be performed.

Alive and dead groups should be compared.

Did you include patients with multiple hematomas?

What are the new points of this study? It has been documented that the higher hematoma volume, deep localization, intraventricular spreading of hematoma, masseffect was significantly associated with a worse outcome.

Comments on the Quality of English Language

fine

Author Response

We thank the editor and reviewers for their careful reviews of our manuscript and insightful comments which helped improve our work. Below is an itemized list of reviewers’ comments and our corresponding response immediately follows. We have highlighted the changes in the revised manuscript and supplemental material documents.

Reviewer #2

C.1. Why the follow-up head CT within 48 hours (or multiple follow-up head CTs) from the ICH symptom onset was done? Potential selection bias (hematoma expansion only in scanned earlier after the onset) should not be ignored.

R: We thank the reviewer for their insightful comments which helped us improve our manuscript. As also addressed in our response to Reviewer 1 questions 1 and 2, in the revised methods (2.1. Patients section), we have clarified the imaging protocol of patients with ICH who are admitted to our center as: “The majority of patients with spontaneous ICH admitted to our center will routinely undergo follow-up CT scans at 6-to-12 and 24-to-48 hours intervals; however, any follow-up imaging is affected by each individual patient risk-factor, clinical course, and treatment plan.”

In the revised manuscript, we have also compared the characteristics of patients who were included in our analysis versus those who had supratentorial ICH but were excluded due to time-frame criteria. The information is added to Results (section 3.1. Patients Characteristics and supplemental Table): “Supplemental Table compared the characteristics of patients who were included in our analysis versus those with supratentorial ICH and follow-up CT who were excluded due to a baseline CT obtained after the first 24 hours from the onset or follow-up CT obtained greater than 48 hours from the onset. The 43 excluded patients fulfilling above criteria had lower systolic blood pressure, less severe symptoms (lower admission NIHSS), higher blood glucose, and better clinical outcome (lower 3-month mRS). They were also more likely to be white and have deep ICH (supplemental Table).”

We agree with reviewer comment that this could have been a source of selection bias and added this to the limitation of our study, stating that “Our inclusion and exclusion criteria induce selection bias as pointed out in the supplemental Table. Patients with milder symptoms and lower blood pressure at presentation were more likely not to undergo head CT scans within 24 hours from the ICH onset or obtain a follow-up CT within the 48-hour post-onset time window and thus were excluded from our study. Our analysis has also inherently excluded patients with massive ICH who succumbed before any follow-up imaging obtained. Thus, our selection criteria have inadvertently omitted individuals exhibiting symptoms at the most severe end of the spectrum.”

C.2. Why systolic blood pressure was almost balanced in HE and non-HE groups (P=0.447)?

R: We agree with the reviewer that higher blood pressure is a known risk factor for hematoma expansion; however, we could not replicate this in our cohort. We have addressed this issue as limitation in our revised manuscript adding that “Although our cohort is relatively large sample size, it could not represent some of the well-established risk factors in ICH patients; for example, there was no significant difference in systolic blood pressure of patients with and without HE, whereas higher blood pressure is a risk-factor for HE”.

C.3. How these patients were treated in this study? Some may argue that different treatment may have effects on final results.

R: We agree with the reviewer and included therapeutic interventions – specifically hematoma evacuation and external ventricular drainage – in the analysis of the revised version. Regarding blood pressure reduction, extraction of precise treatment regimen and therapy response from the medical records is not feasible. None of subjects in our cohort received hemostatic therapy.

In the results section and Tables 1 to 4, we have added the external ventricular drainage and surgical intervention in our univariate analyses. In our multivariate analyses, we included these treatment variables and reported new results in section 3.4, stating that “Given that patients with HE had larger hematomas and worse clinical symptoms at presentation, we evaluated the association of HE with ND, poor outcomes, and mortality after correcting for patients’ age, admission NIHSS, GCS, baseline hematoma volume, intraventricular hemorrhage, external ventricular drainage placement, surgical evacuation, and history of hypertension in multivariable logistic regression. Among patients scanned within the first 3 hours from the ICH onset, HE was an independent predictor of ND with an OR=2.7 (1.6 – 4.5) (p<0.001), poor outcomes with an OR=2.4 (1.2 – 4.6) (p=0.010), and mortality with an OR=2.7 (1.4 – 5.2) (p=0.003), after correcting for other risk-factors and therapeutic interventions. Similarly, among patients scanned 3-to-24 hours after the ICH onset, HE was an independent predictor of ND with an OR=1.4 (1.0 – 2.4) (p=0.043), poor outcomes with an OR=1.7 (1.1 – 3.7) (p=0.037), for poor outcome, and mortality with an OR=3.0 (1.4 – 6.3) (p=0.004), after correcting for other risk-factors and therapeutic interventions. In mediation analysis adjusted for admission NIHSS, GCS, age, baseline hematoma volume, and presence IVH as well as external ventricular drain placement and surgical evacuation, the effects of HE on poor outcome, and death was mediated by ND, with average causal mediation effects of 0.03 (0.01 – 0.05), and 0.03 (0.01 – 0.05), respectively (both p<0.001)

In the limitation of our study, we have added “In addition, aside from surgical evacuation and external ventricular drainage, the details of medical management for blood pressure reduction and strategies for reversal of anticoagulation could not be systematically organized and analyzed in this retrospective analysis, and given the variability of therapeutic options in the absence of universal guidelines. Notably, none of the patients in our cohort received hemostatic agents.”

Q.4. Regarding the rate of subsequent hematoma expansion, did you compare the data with other medical centers?

R: We thank the reviewer for their comments and have compared our results with prior study.

In the introduction, we have stated that “Brott et al reported a 26% rate of >33% hematoma growth among ICH patients scanned within the 1st hour after the initial CT scan, with an additional 12% hematoma growth between the baseline and follow-up scans performed 1 to 20 hours after the baseline scan [5].”

In the discussion, we have added that “These findings confirm prior studies suggesting that HE tends to occur in the very first hours after the ICH onset, although it is still possible that HE occur later on, up to 24 hours after the onset [25,26]. Prior studies have also shown that HE is more common among patients scanned earlier after the ICH onset than those scanned later [4-7]. Kazui et al showed that the frequency of HE progressively declined as the time from onset to the initial scan increases with 36% rate in patients scanned ≤3 hours from the onset, 16% among those scared 3-to-6 hours post-onset, then 15% for 6-to-12 hours, 6% for 12-to-24 hours, and 0% for 24-to-48 hours post-onset time windows [25]. In a recent meta-analysis, the probability of ICH growth, with increasing time from symptom onset to baseline imaging, decreased in a non-linear fashion [7]. The rate of the decline in HE rate was steepest during the 0.5–3 hours after ICH onset [7].”

C.5. How about the relationship between hematoma expansion and short-term outcomes?

R: To address the short-term outcome results, we added the discharge modified Rankin Score as stated in revised method section 2.3 as “For short-term outcome, we used the discharge mRS”. The results are depicted in Tables 1 to 3 of the revised manuscript. Overall, the short-term outcome moves in the same direction as 3-month outcome.

Q.6. Which patients tend to expand at much higher rate in the earliest hours? How about changes of their blood pressure in this study?

R: We have applied multivariate analysis to identify independent predictors of HE and added the results to section 3.2 stating that “In subgroup analysis, in both cohorts of 304 patients scanned within the first 3 hours as well as 263 who were scanned 3-to-24 hours after the ICH onset, baseline hematoma volumes were the only independent predictors of HE.” Baseline systolic blood pressure was not an independent predictor of HE in either sub-cohort. Please also refer to our answer to Comment #2 by Reviewer #2, regarding the limitation of our analysis in identifying the relationship between blood pressure and hematoma expansion.

Q.7. How about the relationship between hematoma expansion and the localization of intracerebral hematoma? Additional analysis should be performed.

R: In the revised manuscript, we have included deep versus lobar location of ICH in our analysis (Tables 1 to 4). While there was a significant difference in localization of ICH between those with and without hematoma expansion (Table 1), the location of hemorrhage was not significantly different based on neurological deterioration (Table 2), poor outcome (Table 3), or mortality (Table 4). We also added that the definition of hematoma expansion (based on percentage versus absolute increase in size) affects the reported ratio of expansion based on the hemorrhage location (Roh et al. Neurocrit Care. 2019; 31(1): 40–45). We have addressed this issue in the discussion as well.

Q.8. Alive and dead groups should be compared.

In the revised manuscript, we have added Tables 3 and 4, comparing the characteristics of patients based on 3-month outcome and mortality. In the results section 3.4, we have added “Table 3 summarizes the clinical characteristics of patients with favorable versus poor outcome at 3-month follow-up, and Table 4 summarizes the clinical characteristics of patients who survived versus those who deceased”

C.9. Did you include patients with multiple hematomas?

R: We have clarified this issue in the revised manuscript. In methods section 2.2, we added that “In patients with multiple ICH lesions, the collective volume of all hemorrhagic lesions was calculated.” In the results section 3.1, we have clarified that “None of the patients had bilateral acute ICH.”

Q.10. What are the new points of this study? It has been documented that the higher hematoma volume, deep localization, intraventricular spreading of hematoma, mass effect was significantly associated with a worse outcome.

R: In the introduction of our manuscript, we have stated the existing knowledge gap and the main goals of our study as “…it remains to be determined whether the prognostic correlates of HE would also change over time after the ICH onset”. We also stated that “…, there remains a knowledge gap regarding whether there is a difference in the association of HE with ND as the rate of hematoma growth decreases over time after the ICH onset.”

In the revised manuscript, we have highlighted the contributions of our work to existing literation, stating that “Our findings suggest that preventing HE — whether in patients presenting more than three hours after the ICH onset or with mild symptoms — has the potential to improve outcomes and reduce mortality.

Our study has several strengths that contribute to its clinical relevance. Firstly, we leveraged a relatively large sample size from a tertiary care stroke center. This leads to the statistical power of our findings and improves the generalizability of our results. Combination of prospective data registry and retrospective review of medical record provided comprehensive information regarding the clinical course during admission and medium-term functional outcome of patients, allowing us to evaluate ND and 3-month outcomes. Additionally, we conducted mediation analysis to determine the underlying mechanisms influencing the relationship between HE and ND with morbidity and mortality. Finally, our multivariate analyses were adjusted for other baseline predictors of outcome and death among ICH patients as well as therapeutic interventions including surgical evacuation and external ventricular drainage.”

Round 2

Reviewer 2 Report

Comments and Suggestions for Authors

The authors have addressed my previous concerns.

Comments on the Quality of English Language

fine

Author Response

We thank the reviewers for their careful reviews of our manuscript and insightful comments which helped improve our work